# Psychometric Properties of the Brazilian Version of GOHAI among Community-Dwelling Elderly People

**DOI:** 10.3390/ijerph192214725

**Published:** 2022-11-09

**Authors:** Mario Vianna Vettore, Maria Augusta Bessa Rebelo, Janete Maria Rebelo Vieira, Evangeline Maria Cardoso, Dina Birman, Anna Thereza Thomé Leão

**Affiliations:** 1Department of Health and Nursing Sciences, University of Agder (UiA), Universitetsveien 25, 4630 Kristiansand, Norway; 2Faculty of Dentistry, Federal University of Amazonas, Av. Ministro Waldemar Pedrosa, 1539, Praça 14 de Janeiro, Manaus 69025-050, Brazil; 3School of Health Sciences, State University of Amazonas, Avenida Carvalho Leal—1777, Manaus 69065-001, Brazil; 4Department of Dental Clinic, Division of Periodontics, School of Dentistry, Universidade Federal do Rio de Janeiro, Rio de Janeiro 21941-617, Brazil

**Keywords:** psychometrics, oral health, quality of life, aged

## Abstract

This study assessed the psychometric properties of the Brazilian version of the Geriatric Oral Health Assessment Index (GOHAI). A representative sample of 613 community-dwelling elderly people aged from 65 to 74 years was selected. Sociodemographic data, GOHAI and self-perceived oral health measures were collected. Dental clinical measures were obtained through oral examinations. The dimensional structure and adequacy of components were assessed using Confirmatory Factor Analysis (CFA), inter-item correlations and item–scale correlations. Reliability was evaluated by internal consistency and Intraclass Correlation Coefficients. Correlations between GOHAI scores and self-reported oral health measures were conducted to assess convergent validity. The relationship between dental clinical measures and GOHAI was tested through Poisson Regression to examine discriminant validity. The link between GOHAI items and dimensions was supported by CFA. Item 12 showed a poor factor loading. The inter-item correlations varied from 0.047 to 0.442, and item–scale correlations ranged from 0.305 to 0.612. Cronbach’s alpha was 0.704. The test–retest correlation for GOHAI was 0.882. GOHAI scores were correlated by self-rated oral health measures. Poor dental clinical measures were associated with GOHAI. The Brazilian version of GOHAI showed adequate psychometric properties. However, the weak dimensional structure of GOHAI suggests the need to perform cross-cultural adaptation of GOHAI for Brazilian elderly people.

## 1. Introduction

The process of population ageing is a global phenomenon initiated in the 1970s that is expected to continue throughout the 21st century. By 2030, the world elderly population will be around 1 billion people, which means that 1 in every 8 of the earth’s inhabitants are expected to be aged 65 years and older [1]. Furthermore, it is anticipated that the number of elderly people will exceed the number of young by 2050 if the demographic forecast trends are confirmed [2]. The arrival of an ageing population imposes several challenges to public health systems with great impacts on health services utilization due to an increase in the global prevalence of chronic diseases and dementia [2,3]. The demographic transition has also been influencing the global trends in oral health, as elderly people are retaining their natural teeth for longer with a consequent increase in dental treatment needs [4].

Contemporary approaches in oral health assessment and healthcare needs acknowledge the limitations of using normative methods, such as dental clinical indices, in the evaluation of individuals’ oral health. Thus, the importance of individuals’ own evaluation of their oral health status through self-reported measurements has gained attention over the last decades [5,6]. The subjective oral health assessment also considers the broad perspective of the WHO’s multidimensional definition of health through incorporating the central elements of the health-related quality of life concept, including functioning, psychological and social aspects, and pain or discomfort [5,6]. Oral health-related quality of life (OHRQoL) is a multidimensional construct referring to the individuals’ perception of the oral symptoms, functional status, psychological impacts, and social well-being affected by oral disorders [6].

The value of assessing OHRQoL as a comprehensive measure of oral health as opposed to dental disease and oral illness became clear and worthwhile in dental research and policy decisions related to oral health by contextualizing the impact of dental problems on quality of life and well-being at the individual and population levels [7]. Nonetheless, the comprehensive conceptualization of OHRQoL and the complexity of OHRQoL indicators indicated the need for valid and reliable instruments in order to reflect the multidimensional aspects of the OHRQoL construct [6].

Several generic and specific OHRQoL instruments have been developed over the last decades and have been used to assess population needs in health services research, individual and contextual determinants of oral health and responsiveness of oral health status to different interventions [5]. Among the age-specific OHRQoL instruments, the General Oral Health Assessment Index Questionnaire (GOHAI) was developed by Atchison and Dolan in the USA to assess oral health in older adult populations aged 65 years and older [8]. The 12 items of the GOHAI questionnaire were selected to express elderly people´s oral health problems according to three dimensions: (1) physical function, (2) psychosocial function, and (3) pain or discomfort [8,9].

The usefulness of the GOHAI instrument has been recognized as an important tool to assess OHRQoL in elderly people because of the unique physical and psychological characteristics of this age group related to their oral health, including higher prevalence of partial or total tooth loss (edentulism) and poor perception of dental disease such as dental caries and periodontal disease compared to younger age groups [10]. Therefore, the GOHAI instrument was translated and validated into different countries and languages and has been widely used in clinical and epidemiologic research [11,12,13,14,15,16].

The GOHAI Instrument was not cross-culturally adapted into Portuguese. However, the translated version of GOHAI into Portuguese was used to assess the relationship of clinical characteristics, subjective measures, and sociodemographic factors with self-perceived oral health in elderly people attending a rehabilitation centre in a middle-sized city in Brazil [17,18] (Table 1). Previous research has also used the Portuguese version of the GOHAI to evaluate the oral health and self-perceived oral health of elderly people with and without access to dental treatment [19] and to examine the association of demographic, socioeconomic and dental clinical measures with OHRQoL in elderly people [20]. In addition, the validity of GOHAI was investigated among edentulous Brazilian elderly people attending a dental clinic in a public university in Brazil [21]. However, no study has assessed the dimensional structure, reliability, and validity of the GOHAI questionnaire amongst Brazilian elderly people. The aim of this study was to evaluate the psychometric properties of the translated version of GOHAI in a representative sample of community-dwelling elderly people.

## 2. Materials and Methods

### 2.1. Study Population and Ethics

A sample of 613 elderly people was selected in the city of Manaus, Brazil. The participants were recruited in their households using a stratified random clustered sampling to obtain a representative sample of the 27,853 elderly residents living in Manaus. The sample was obtained from census tracts according to the proportion of the population distributed in the five administrative regions of the city: Centre-South, Midwest, East, North, West and South. Further details of the sampling methodology were published elsewhere [22]. All residents aged between 65 and 74 years in good health condition to be dentally examined and presenting adequate cognitive function according to a minimum score of the Verbal Fluency Test were considered eligible to participate [23].

The main study was conducted in 2007 to estimate the prevalence of edentulism in the city of Manaus. The sample size was originally estimated as 807 individuals considering the proportion of 53% of edentulism in the Northern Region of the country [24], with a margin of error of 10%, 95% of significance, design effect of 2 and a non-response rate of 20%.

Of the 810 elderly people screened in the survey, 766 agreed to participate (response rate = 94.6%). Eight-four elderly people could not be examined because of their poor health status and a further 12 were excluded as they were considered as having cognitive impairment to respond the questionnaire. Fifty-four participants with incomplete data were excluded from the analysis, resulting in a final analytic sample of 613 elderly people.

The present study was approved by the Ethics Committee of the Federal University of Amazonas (Protocol No. 0234.0.115.000-07). Before undergoing the interview and dental clinical examinations, all participants signed the written informed consent. Detailed information about the aim of the study and data collection procedures was also provided.

### 2.2. The GOHAI Instrument

The GOHAI is composed of 12 items grouped in three dimensions: (1) physical function, including eating, talking and swallowing (items #1, #2, #3, #4); (2) psychosocial aspects, including worry or concern about oral health, dissatisfaction with appearance, self-consciousness about oral health, and avoidance of social contacts because of oral problems (items #6, #7, #9, #10, #11); and (3) pain or discomfort, including the use of medications to relieve pain or discomfort related to oral health problems (items #5, #8, #12) [8,9]. The items assess the impact of oral health conditions on everyday life over a 3-month reference period and are responded to using a six-point Likert scale: 0 = never, 1 = seldom, 2 = sometimes, 3 = often, 4 = very often and 5 = always. Originally, the responses to nine items—item #1 “limit food due to dental problems”, item #2 “trouble biting and chewing”, item #4 “prevented from speaking”, item #6 “limited contacts with people”, item #8 “used medication”, item #9 “worried about teeth”, item #10 “nervous due to teeth”, item #11 “uncomfortable eating with people”, and item #12 “sensitive to temperature”—are reversed before calculating the GOHAI score [8,9].

The GOHAI score is obtained by summing the scores of the individual items, resulting in the “additive score” (ADD-GOHAI) that ranges from 0 to 60 with the higher ADD-GOHAI scores indicating better OHRQoL. The “simple count score” (SC-GOHAI) is recorded by adding one point for each item answered with “sometimes”, “often” or “always”, and may vary from 0 to 12 [11,13,14,16]. Higher SC-GOHAI scores suggested poor OHRQoL.

### 2.3. Brazilian Version of GOHAI

The GOHAI items were translated into the Portuguese language by Brazilian researchers [18,19]. The structure of the instrument considering 12 items representing the three theoretical dimensions was maintained. However, eleven items were worded positively and only the score of item #7 “pleased with appearance” should be reversed before computing the GOHAI scores. In addition, an alternative scoring was adopted as the six response categories were replaced by a 3-point Likert scale with the following options: 1 = always/often, 2 = sometimes/seldom, 3 = never (Table 1). The ADD-GOHAI varies from 12 to 36. The SC-GOHAI is a count of the items with the responses “sometimes” and “always” and ranges from 0 to 12.

### 2.4. Data Collection and Measures

The participants were interviewed and examined by a single and previously calibrated examiner in their households. The examiner read the items of all questionnaires aloud during the interview, including the GOHAI items. An artificial head light, oral plain mirror No. 5 (Duflex) and CPI periodontal probe (Stainless) were used during the oral examinations following biosafety rules. Test–retest reliability of the clinical measurements was determined in twenty older people attending a public community centre over 7 days. The kappa coefficient was 0.97 and 1 for DMFT and need for dentures, respectively.

Sociodemographic data included age, sex (male; female) and educational level (0–4 years; 5–8 years and ≥9 years). Subjective measures were dental pain scale assessed using a 4-point Verbal Rating Scale ranging from “no pain” to “a lot of pain” [25]. Global oral health rating questions were self-perceived measures including questions about oral health assessment, treatment needs, dental appearance, masticatory function, speaking function and social function. The five first items were answered on a 4-point Likert scale, ranging from “very poor” to “very good”. The response options for social function varied from “does not affect at all” to “affect a lot”.

Dental clinical measures were registered following the criteria proposed by the World Health Organization [26]. The need for total dental prosthesis was registered when the participant had no natural teeth in the dental arch. Participants whose existing dentures required replacement due to problems in retention, stability, fixation, or aesthetics were also deemed to have denture needs. The total dental prosthesis need was recorded as “0 = no need of total dental prosthesis’’, “1 = upper or lower need of total dental prosthesis”, and “2 = need of upper and lower total dental prostheses”. The number of decayed teeth was recorded according to the component of the decayed, missing and filled teeth index (DMFT) [26]. Functional dentition was measured according to the number of natural teeth using the following categories “No” (<20 teeth) or “Yes” (≥20 teeth) [27]. The community periodontal index (CPI) including measures of bleeding on probing, dental calculus and periodontal probing were registered for each sextant [28]. The latter was not included in the analyses due to very low frequency of periodontal pockets ≥ 4 mm (2.7%).

### 2.5. Statistical Analysis

#### 2.5.1. Descriptive

Proportions (%) of each of the GOHAI items and mean (SD) of the ADD-GOHAI (additive) score and the SC-GOHAI (simple count) scores and GOHAI dimensions were calculated.

#### 2.5.2. Dimensional Structure and Adequacy of Components

The dimensional structure and adequacy of components of the GOHAI were evaluated through Confirmatory Factor Analysis (CFA), inter-item correlations and correlations between item scores and the GOHAI score of the related dimensions (subscales). The CFA model tested the factor loadings of GOHAI items (indicators) and the multidimensionality of the instrument according to the three hypothetical dimensions: (1) physical function, (2) psychosocial function, and (3) pain or discomfort represented as latent variables. Loading coefficients > 0.30 were considered acceptable. The maximum likelihood estimation method and bootstrapping were used to estimated standardized betas (Factor loadings) and 95% Confidence Intervals (CI) using SPSS AMOS 22.0. Factor loadings > 0.3 were considered salient [29]. The bootstrapping procedure was performed through re-sampling 900 samples from the original data set to derive less biased standard errors and 95% CI bootstrap percentiles. The adequacy of the CFA model was evaluated using the following fit indexes and threshold values: χ^2^/df < 3.0, comparative fit index (CFI) ≥ 0.90, goodness-of-fit index (GFI) ≥ 0.90, standardized root-mean-square residual (SRMR) ≤ 0.08, and root-mean-square error of approximation (RMSEA) ≤ 0.06 [30].

Pearson coefficient was used to estimate inter-item correlations as well as the correlations between item scores and the GOHAI score of each dimension. The average inter-item correlation ideally should be between 0.2 and 0.5 [31].

#### 2.5.3. Reliability

Reliability was assessed by measuring internal consistency and stability. Internal consistency was measured through Cronbach’s alphas of the GOHAI and GOHAI dimensions with 95% CI [32]. Alpha values of 0.7 to 0.8 are regarded as satisfactory [33]. Correlations between item scores and the overall GOHAI score were assessed using the corrected item–total score correlation (Spearman’s rank correlation coefficient). In addition, the GOHAI Cronbach’s alpha if an item was deleted was calculated.

Stability was assessed by measuring test–retest reliability through Intraclass Correlation Coefficients (ICCs) for the individual items, GOHAI total score and dimensions using ADD-score and SC-score. A second interview was conducted among 74 participants randomly selected who completed the GOHAI questionnaire twice with a one-week interval. It was assumed that no large changes in their dental status or oral health occurred during this time interval and, therefore, high stability should be observed. ICC values > 0.75 were considered indicative for excellent stability, 0.40–0.75 for fair to good, and <0.40 for poor stability [34].

#### 2.5.4. Validity

Convergent validity encompasses the examination of the degree to which two measures that should measure similar constructs are related. Pearson’s coefficient was employed to assess the correlations of GOHAI and GOHAI dimensions using ADD-score and SC-score with the dental pain scale and self-perceived oral health measures.

Discriminant validity was assessed using multivariate Poisson Regression to obtain coefficients and 95% CIs. The dependent variable was GOHAI ADD-score, and the independent variables were the need of total dental prosthesis, functional dentition, number of decayed teeth, number of sextants with bleeding on probing and number of sextants with calculus. The models were adjusted for age, sex, and educational level.

## 3. Results

### 3.1. Descriptive

The present study included 613 elderly people (mean age 69.27 years, SD = 3.01). Of them, 69.5% were females and 59.1% had up to 4 years of schooling. Fifty-four percent of the participants were edentulous (missing all natural teeth). The need for upper and lower total dental prosthesis was observed in 28.9% of the participants and 50.4% did not need total dental prosthesis. Only 2.3% of the sample had functional dentition (≥20 teeth), and the mean of decayed teeth was 0.54 (SD = 1.58).

The original English version and the Portuguese version of the GOHAI items and distribution of responses in the studied sample are shown in Table 1. The responses to physical functional items revealed that 9.1% and 23.8% of the participants informed of limitations (sometimes or always) in the kinds or amounts of food eaten (item 1) and problems in biting or chewing (item 2), respectively. Most participants reported never having problems to swallow comfortably (item 3) (94.1%) and to speak (item 4) (81.4%). Pain and discomfort questions indicated that 17.4% of the participants felt discomfort (sometimes or always) when eating (item 5), 5.9% reported the use of medication to relieve oral pain (item 8), 24.1% reported sensitivity with their teeth (item 12). The psychosocial function’s answers suggested that most people never limit contacts with people because of their teeth or dentures conditions (item 6) (96.9%) and were happy with the appearance of their teeth or dentures (item 7) (75%). Furthermore, 11.3% were (sometimes or always) worried or concerned about their oral health (item 9), 4.9% were nervous because of problems in their mouth (item 10), and 8.6% felt uncomfortable eating in front of people (item 11) (Table 1).

The means for the total ADD-GOHAI (additive score) and total SC-GOHAI (simple count) scores were 33.90 (SD = 2.70) and 1.58 (SD = 1.96), respectively. ADD-GOHAI physical function, psychosocial function, and pain or discomfort dimensions average scores were 11.30 (SD = 1.19), 14.27 (SD = 1.26) and 8.34 (SD = 1.00). The corresponding figures for the dimensions of SC-GOHAI (simple count) were 0.57 (SD = 0.90), 0.53 (SD = 0.88) and 0.48 (SD = 0.69), respectively.

### 3.2. Dimensional Structure and Adequacy of Components

The fit indices of the CFA model supported the relationships between the GOHAI items and the hypothetical dimensions according to the following values: χ2/df = 2.205, CFI = 0.942, GFI = 0.971, SRMR = 0.039 and RMSEA = 0.044. The dimension physical function was confirmed by the items “1. Limit foods” (β = 0.465), “2. Trouble biting, chewing” (β = 0.630), “3. Swallow comfortably” (β = 0.377), “4. Trouble speaking” (β = 0.545). The items “6. Limit social contacts” (β = 0.503), “7. Pleased with appearance” (β = 0.432), “9. Worry/concern” (β = 0.374), “10. Nervous/self-conscious” (β = 0.352) and “11. Uncomfortable eating with people” (β = 0.545) confirmed the dimension psychological function. The items that confirmed the pain and discomfort dimension were “5. Eat without discomfort” (β = 0.639), “8. Use of medication” (β = 0.402) “12. Teeth or gums sensitive” (β = 0.185) (Figure 1).

The GOHAI inter-item correlations varied from 0.047 (between item 10 and item 12) to 0.442 (between item 2 and item 5). GOHAI item–dimensions correlations ranged between 0.162 (between item 12 and the physical function dimension) and 0.809 (between item 12 and the pain or discomfort dimension) (Table 2).

### 3.3. Reliability

Cronbach’s alpha for the GOHAI was 0.704 (95% CI = 0.668–0.737). Cronbach’s alpha for the physical function dimension was 0.568 (95% CI = 0.509–0.621), the psychosocial function dimension was 0.495 (95% CI = 0.429–0.556), and the pain or discomfort dimension was 0.240 (95% CI = 0.129–0.338). Correlations between item scores and the overall GOHAI score ranged from 0.305 to 0.612 (Table 3). The removal of item 12 was the only aspect that increased the GOHAI Cronbach’s alpha (from 0.704 to 0.733). The test–retest correlation for single items varied from 0.573 (item 1) to 1.000 (item 6). The test–retest correlation for GOHAI ADD-scores (ICC = 0.882) and GOHAI SC-scores (ICC = 0.881) suggested excellent stability. The GOHAI dimensions test–retest correlation ranged from 0.713 (pain or discomfort SC-score) to 0.955 (psychosocial function SC-score) (Table 3).

### 3.4. Validity

The convergent validity of GOHAI scores was assessed through the correlation of the GOHAI total score and GOHAI dimension scores with dental pain scale and self-rated oral health measures. Low GOHAI ADD-scores, low ADD-scores dimensions, high GOHAI SC-scores, and high SC-scores dimensions were significantly correlated with low dental pain and low self-perceived social function. Significant correlations were observed between high GOHAI ADD-scores, high ADD-scores dimensions, low GOHAI SC-scores and low SC-scores dimensions, and self-rated oral health, self-perceived treatment needs, self-perceived dental appearance, self-perceived masticatory function, and self-perceived speaking function (Table 4).

Poisson Regression analysis assessed the discriminant validity of GOHAI through estimating the association of dental clinical measures with ADD-GOHAI overall scores and ADD-GOHAI dimension scores. Participants with upper and lower total dental prosthesis needs and more decayed teeth had lower scores of ADD-GOHAI overall scores and ADD-GOHAI physical function and psychosocial function scores. Participants with functional dentition (≥20 teeth) had higher scores of ADD-GOHAI overall scores and ADD-GOHAI physical function and psychosocial function scores. A greater number of sextants with dental calculus was associated with higher GOHAI scores and GOHAI dimension scores (Table 5).

## 4. Discussion

Over recent decades, OHRQoL assessments have been included in observational studies and clinical trials to investigate the predictors of OHRQoL as well as the effectiveness of interventions with the goal of improving oral health [5]. The progressive use of OHRQoL measures in dental research has been supported by the development of numerous instruments that began to be used outside of the populations where they were developed [7]. In light of the fact that peoples’ perceptions of their oral health are culturally specific, it is necessary to confirm the cross-cultural equivalency of the instruments assessing self-reported health measures, including OHRQoL, to obtain valid and reliable measures before carrying out between-population comparisons [35]. In this study, the psychometric properties of the GOHAI version previously translated for Brazilian elderly people was assessed among community-dwelling elderly people living in a large city [17,18]. To date, there is a dearth of studies evaluating the properties of GOHAI in Portuguese-speaking populations [21]. Previous research involving edentulous elderly Brazilians concluded that GOHAI demonstrated good construct validity [21]. However, the use of a convenience sample in a small sample size, and the specific oral health status of the participants (edentulous) of the former study, suggested the need for further assessment of GOHAI properties [21].

The GOHAI was developed to measure oral functional problems, psychosocial impacts and pain related to dental problems in the elderly population. The instrument is considered suitable to assess patient-reported outcomes in clinical and epidemiological research in this age group. GOHAI is grounded by three assumptions. First, oral health status is measurable considering the individual self-assessment of the elderly person. Second, variations of oral health levels among elderly people can be detected using self-reported measures. Third, dental status predict the perception of oral health among elderly [8,9]. GOHAI was initially developed in the English language for elderly Americans, and the translation and cross-cultural equivalence of the instrument has been mainly conducted into other European languages [11,13,14,15,16]. The noteworthy cultural differences as well as disparities in the oral health status and in the use of dental services of elderly people between European countries and Latin America countries reinforce the need for appropriate assessment of the validity of GOHAI for each country [4].

The frequency distributions of GOHAI items in this study reveal that the sample tended towards providing positive answers. This was particularly so for the items “3. Swallow comfortably”, “6. Limit social contacts”, and “10. Nervous/self-conscious”, suggesting that the oral health problems of the participants were not major obstacles to perform daily activities. The participants of this study reported that sensitivity was the major problem as nearly 16% of them informed of sensitivity in their teeth or gums all the time. These findings are possibly related to the sampling method and the eligibility criteria used in this study as the participants were community-dwelling elderly people younger than 75 years old. Overall, the Brazilian version of the GOHAI used in this study showed good reliability and validity, though some of the pre-established criteria used to assess the psychometric properties were not met.

The changes in the Brazilian version of the GOHAI were not limited to the wording of the original scale items. The original scoring system using a 6-point Likert scale was replaced by a 3-point Likert scale in the Brazilian version [17,18]. Consequently, the range of GOHAI score reduced from 12–60 to 12–36. The number of response categories of the original GOHAI used in previous studies is not consistent [9]. Alternative scoring of GOHAI included a Likert-type scale with 3 categories (response categories 1–3, GOHAI range 12–36) and 5 categories (response categories 1–5, GOHAI range 12–60) [11,15,16]. The rescoring of GOHAI using different response categories was proposed to allow direct comparisons among studies adopting distinct categories [9]. The possible implication of using a 3-point Likert scale in the present study is the reduction in the convergent and discriminant validity of the scale. However, GOHAI and its dimensions were significantly correlated with dental pain scale and self-perceived oral health measures. In addition, distinct dental clinical measures, including the need of total dental prosthesis, functional dentition, number of decayed teeth, and periodontal measures were associated with GOHAI and GOHAI dimension scores. The modification in the number of response categories on the Likert-type scale was also used in other versions of GOHAI [9]. Another relevant change was the shift in the direction of the wording in several items. The original GOHAI has nine items worded positively, whereas only one item was worded in favour of better quality of life in the Brazilian version [8,17,18]. These modifications did not seem to impact on the validity of the questionnaire.

The dimensional structure and adequacy of components of the Brazilian version of GOHAI may have been affected by the changes in the wording in some items. Our findings suggest that item 12 does not seem to have been translated adequately, because the factor loading obtained in the CFA was lower than 0.3 [29]. Moreover, item 12 showed weak correlations (inter-item correlation < 0.2) with seven items of the GOHAI scale and Cronbach’s alpha increased only when item 12 was removed. The original version of item 12 asks “How often were your teeth or gums sensitive to hot, cold or sweet foods?”. However, in the Brazilian version of item 12, the question only refers to “sensitivity of teeth or gums to liquid foods” [17,18]. The original version of items 7, 10 and 11 refer to impact on functional problems and psychosocial function on “teeth, gum or dentures”. However, these items just mention “mouth” in the Brazilian version. The lack of specification in these items might have influenced the participants’ responses. This is because more than 50% of the sample participants were edentulous and nearly 30% need upper and lower dentures. Although the factor loadings of items 7, 10 and 11 were acceptable, problems in the translation may have introduced information bias and affected some of the overall psychometric measures of GOHAI. The abovementioned findings may have occurred because the cross-cultural adaptation of GOHAI for Brazilian elderly people was not conducted yet. Instead, the translated version of GOHAI into Portuguese was used in previous research but the psychometric characteristics of the instrument was not evaluated until now [17,18,19,20,21].

The internal consistency of the Brazilian version of GOHAI can be considered adequate as the Cronbach’s alpha in the studied sample (α = 0.704) was slightly higher than 0.70. This finding was lower than the Cronbach’s alpha obtained when GOHAI was developed (α = 0.79). Previous studies reported higher Cronbach’s alpha when the reliability of GOHAI was assessed in other countries, including Saudi Arabia (α = 0.93) [12], Germany (α = 0.92) [13], Spain (α = 0.87) [16], The Netherlands (α = 0.86) [14], and Sweden (α = 0.86) [11]. In the present study, Cronbach’s alpha of GOHAI dimensions were below 0.60, which indicates low reliability. Few studies evaluating the psychometric properties of GOHAI in other countries reported the Cronbach’s alpha of GOHAI dimensions [14]. The Brazilian version of GOHAI showed excellent stability for the total score (ICC = 0.822) and GOHAI dimensions, except for the pain or discomfort dimension when measured using the simple count (ICC = 0.713). The temporal reliability of GOHAI in other countries was usually very good, varying from 0.64 [11] to 0.95 [12]. The stability observed in this study was similar to those reported in German (ICC = 0.84) [13] and Dutch (ICC = 0.88) [14] versions of GOHAI.

The interpretation of the present findings must be cautious and should not be generalised to elderly people aged 75 and older and those in poor general health status such as those with dementia and hospitalized patients. In addition, our findings should not be applied to institutionalized and frail elderly people, including those living in long-term care homes. Elders living in institutions and frail individuals suffer worse oral health conditions and have poor OHRQoL than those residents in the community [4,36].

The importance of measuring OHRQoL is well established and relies on the shift from traditional clinical measures to assessing the different functional, emotional, and social aspects of oral health to establish realistic treatment objectives and outcomes [5]. Yet, OHRQoL measures for elderly people might be considered narrow since they essentially evaluate the impact of oral health conditions [6]. The GOHAI is particularly limited compared to other OHRQoL measures due to a greater emphasis on functional limitations, pain and discomfort and less discriminant validity when compared with OHIP-14 [37]. Recently, new questionnaires have been developed to assess the elderly’s perception of the chronicity of their underlying oral disease and should be considered in future studies, such as the Illness Perception Questionnaire Revised for Dental Use in Older/Elder Adults (IPQ-RDE) [38].

## 5. Conclusions

The Brazilian version of GOHAI showed adequate internal consistency and stability as well as satisfactory convergent and discriminant validities. However, the detected problems in the dimensional structure and adequacy of components of GOHAI in this study suggest that cross-cultural adaptation of the instrument is necessary, including the semantic equivalence to transfer the meaning of concepts contained in the original instrument to the translated version. The dimensional structure, reliability and validity should be re-assessed among Brazilian elderly people to confirm whether GOHAI can be used to assess OHRQoL in this population group.

## Figures and Tables

**Figure 1 ijerph-19-14725-f001:**
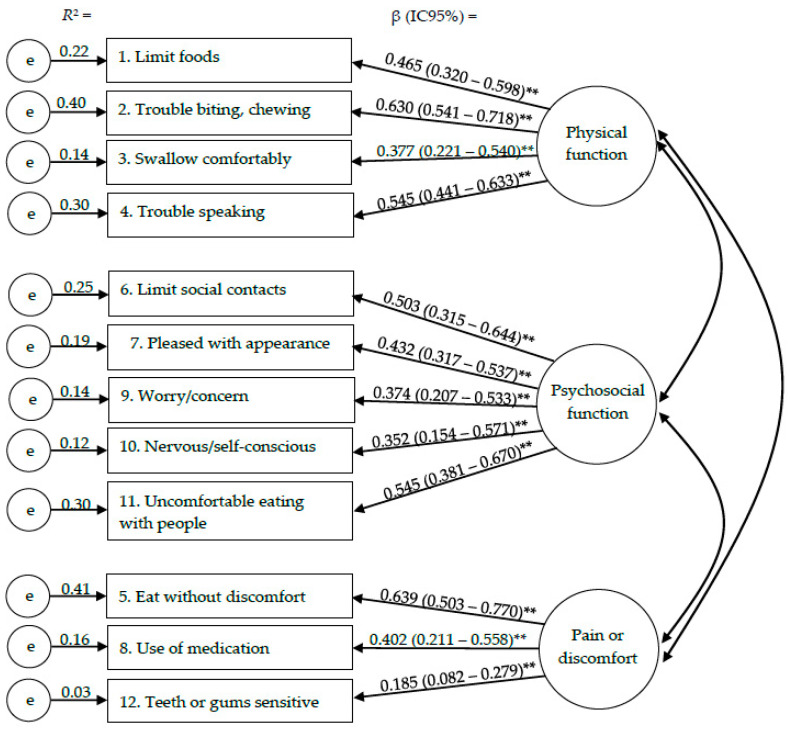
Confirmatory factor analysis of the three-factor and twelve-item measurement model obtained through bootstrap item loadings (standard error/bias-corrected 95% CI). ** *p* < 0.01.

**Table 1 ijerph-19-14725-t001:** GOHAI items and frequency distribution of the responses (*n* = 613).

Item	Dimension	In the Past Three Months (*Nos últimos três meses*)	(1) A (*S*)	(2) S (*AV*)	(3) N (*N*)
1	Physical function	How often did you limit the kinds or amounts of food you eat because of problems with your teeth or denture?	13 (2.1)	43 (7.0)	557 (90.9)
		*Você diminuiu a quantidade de alimentos ou mudou o tipo de alimentação por causa de seus dentes?*			
2	Physical function	How often did you have trouble biting or chewing any kinds of food, such as a firm meat or apples?	33 (5.4)	113 (18.4)	467 (76.2)
		*Você teve problemas para mastigar os alimentos?*			
3	Physical function	How often were you able to swallow comfortably?	6 (1.0)	30 (4.9)	577 (94.1)
		*Você teve dor ou desconforto para engolir alimentos?*			
4	Physical function	How often have your teeth or dentures prevented you from speaking the way you wanted?	25 (4.1)	89 (14.5)	499 (81.4)
		*Você mudou o jeito de falar por causa dos problemas em sua boca?*			
5	Pain or discomfort	How often were you able to eat anything without feeling discomfort?	16 (2.6)	91 (14.8)	506 (82.5)
		*Você sentiu algum desconforto ao comer algum alimento?*			
6	Psychosocial function	How often did you limit contacts with people because of the condition of your teeth or dentures?	1 (0.2)	18 (2.9)	594 (96.9)
		*Você deixou de se encontrar com outras pessoas por causa de sua boca?*			
7	Psychosocial function	How often were you pleased or happy with the appearance of your teeth, gums or dentures?	460 (75.0)	71 (11.6)	82 (13.4)
		*Você se sentiu satisfeito ou feliz com a aparência de sua boca?*			
8	Pain or discomfort	How often did you use medication to relieve pain or discomfort around your mouth?	3 (0.5)	33 (5.4)	577 (94.1)
		*Você teve que tomar remédio para passar a dor ou desconforto de sua boca?*			
9	Psychosocial function	How often were you worried or concerned about the problems with your teeth, gums or dentures?	19 (3.1)	50 (8.2)	544 (88.7)
		*Você teve algum problema na boca que o deixou preocupado?*			
10	Psychosocial function	How often did you feel nervous or self-conscious because of problems with your teeth, gums or dentures?	4 (0.7)	26 (4.2)	583 (95.1)
		*Você chegou a se sentir nervoso por causa de problemas na boca?*			
11	Psychosocial function	How often did you feel uncomfortable eating in front of people because of problems with your teeth or dentures?	20 (3.3)	33 (5.4)	560 (91.4)
		*Você evitou comer junto com outras pessoas por causa dos problemas na sua boca?*			
12	Pain or discomfort	How often were your teeth or gums sensitive to hot, cold or sweet foods?	96 (15.7)	52 (8.5)	465 (75.9)
		*Você sentiu seus dentes ou a gengiva ficarem sensíveis a alimentos líquidos?*			

A/*S*: Always/*Sempre;* S/*AV*: Sometimes/*Algumas vezes*; N/*N*: Never/*Nunca.* Item 7 scores were reversed before summing the GOHAI scores. The GOHAI total score may range from 12 to 36 and a greater score means better OHRQoL.

**Table 2 ijerph-19-14725-t002:** Inter-item correlations for the GOHAI and item–GOHAI dimensions correlations (Pearson coefficient).

Item	1	2	3	4	5	6	7	8	9	10	11	12
1. Limit foods	1											
2. Trouble biting, chewing	0.330 **	1										
3. Swallow comfortably	0.259 **	0.169 **	1									
4. Trouble speaking	0.225 **	0.347 **	0.195 **	1								
5. Eat without discomfort	0.273 **	0.442 **	0.289 **	0.413 **	1							
6. Limit social contacts	0.272 **	0.268 **	0.201 **	0.215 **	0.227 **	1						
7. Pleased with appearance	0.162 **	0.284 **	0.111 **	0.216 **	0.229 **	0.201 **	1					
8. Use of medication	0.191 **	0.283 **	0.136 **	0.210 **	0.271 **	0.124 **	0.149 **	1				
9. Worry/concern	0.162 **	0.185 **	0.039	0.165 **	0.208 **	0.165 **	0.156 **	0.279 **	1			
10. Nervous/self-conscious	0.121 **	0.115 **	0.147 **	0.155 **	0.224 **	0.235 **	0.116 **	0.238 **	0.417 **	1		
11. Uncomfortable eating with people	0.218 **	0.287 **	0.219 **	0.238 **	0.295 **	0.289 **	0.240 **	0.155 **	0.216 **	0.169 **	1	
12. Teeth or gums sensitive	0.096 *	0.136 **	0.048	0.133 **	0.048	0.083 *	0.085 *	0.154 **	0.122 **	0.047	0.069	1
*GOHAI dimensions*												
Physical function	0.641 **	0.772 **	0.489 **	0.710 **	0.542 *	0.354 **	0.308 **	0.317 **	0.225 **	0.205 **	0.360 **	0.162 **
Psychosocial function	0.283 **	0.380 **	0.207 **	0.319 **	0.375 **	0.460 **	0.748 **	0.297 **	0.609 **	0.501 **	0.613 **	0.134 **
Pain or discomfort	0.249 **	0.381 **	0.205 **	0.345 **	0.570 **	0.200 **	0.208 **	0.504 **	0.261 **	0.201 **	0.229 **	0.809 **

* *p* < 0.05; ** *p* < 0.01.

**Table 3 ijerph-19-14725-t003:** Item–scale correlation, Cronbach’s alpha if item deleted and test–retest-correlation for single items and GOHAI ADD-scores, GOHAI SC-scores.

	Internal Consistency		Stability
Item	Item–Scale Correlation ^a^	Cronbach Alpha If Item Deleted	Test–Retest Correlation ^b^
1. Limit foods	0.369 **	0.680	0.536 **
2. Trouble biting, chewing	0.612 **	0.655	0.690 **
3. Swallow comfortably	0.305 **	0.693	0.794 **
4. Trouble speaking	0.526 **	0.668	1.000 **
5. Eat without discomfort	0.538 **	0.661	0.717 **
6. Limit social contacts	0.272 **	0.692	1.000 **
7. Pleased with appearance	0.560 **	0.694	0.917 **
8. Use of medication	0.326 **	0.686	0.833 **
9. Worry/concern	0.410 **	0.685	0.981 **
10. Nervous/self-conscious	0.292 **	0.693	0.646 **
11. Uncomfortable eating with people	0.385 **	0.677	0.864 **
12. Teeth or gums sensitive	0.433 **	0.733	0.881 **
GOHAI			
GOHAI total			
ADD-score	–	–	0.822 **
SC-score	–	–	0.881 **
Physical function			
ADD-score	–	–	0.776 **
SC-score	–	–	0.760 **
Psychosocial function			
ADD-score	–	–	0.946 **
SC-score	–	–	0.955 **
Pain or discomfort			
ADD-score	–	–	0.859 **
SC-score	–	–	0.713 **

^a^ Spearman’s correlation coefficient. ^b^ Intraclass correlation coefficient. ADD-GOHAI (additive score of GOHAI) is a sum score, ranging from 12 to 60 (high scores indicate few problems). SC-GOHAI (additive score of GOHAI) is a count of the items with the responses “sometimes”, “often” and “always” and ranges from 0 to 12 (high scores indicates poor oral health). Item 7 scores were reversed before calculating the ADD-GOHAI and SC-GOHAI. ** *p* < 0.01.

**Table 4 ijerph-19-14725-t004:** Correlation of total score and dimension scores of GOHAI (ADD-GOHAI) with dental pain scale, self-rated oral health, self-perceived treatment needs, self-perceived dental appearance, self-perceived masticatory function, self-perceived speaking function and self-perceived social function.

	Total Score		Physical Function		Psychosocial Function		Pain or Discomfort	
	Additive Score	Simple Count	Additive Score	Simple Count	Additive Score	Simple Count	Additive Score	Simple Count
Dental pain scale	−0.259 **	0.277 **	−0.171 **	0.187 **	−0.234 **	0.236 **	−0.202 **	0.243 **
Self-rated oral health	0.326 **	−0.314 **	0.162 **	−0.159 **	0.374 **	−0.348 **	0.225 **	−0.242 **
Self-perceived treatment needs	0.258 **	−0.274 **	0.167 **	−0.180 **	0.293 **	−0.292 **	0.126 **	−0.171 **
Self-perceived dental appearance	0.299 **	−0.295 **	0.229 **	−0.220 **	0.314 **	−0.288 **	0.139 **	−0.183 **
Self-perceived masticatory function	0.439 **	−0.453 **	0.437 **	−0.448 **	0.345 **	−0.326 **	0.241 **	−0.285 **
Self-perceived speaking function	0.367 **	−0.378 **	0.414 **	−0.418 **	0.265 **	−0.255 **	0.160 **	−0.202 **
Self-perceived social function	−0.250 **	0.244 **	−0.174 **	0.162 **	−0.263 **	0.255	−0.131 **	0.155 **

** *p* < 0.001 (Pearson’s correlation).

**Table 5 ijerph-19-14725-t005:** Association of GOHAI total scores (ADD-GOHAI) and GOHAI dimension scores with need of total dental prosthesis, functional dentition (at least 20 teeth), number of decayed teeth, number of sextants with bleeding on probing and number of sextants with dental calculus.

	GOHAI Total Score	GOHAI Physical Function	GOHAI Psychosocial Function	GOHAI Pain or Discomfort
Need of total dental prosthesis				
None	Ref	Ref	Ref	Ref
Upper or lower total dental prosthesis	0.007 (−0.009/0.022)	0.001 (−0.021/0.022)	0.009 (−0.007/0.025)	0.010 (−0.015/0.035)
Upper and lower total dental prosthesis	−0.021 (−0.038/−0.004) *	−0.029 (−0.052/−0.006) *	−0.024 (−0.043/−0.005) *	−0.005 (−0.028/0.019)
Functional dentition				
No (<20 teeth)	Ref	Ref	Ref	Ref
Yes (≥20 teeth)	0.046 (0.016/0.076) **	0.048 (0.024/0.071) **	0.044 (0.001/0.087) *	0.045 (−0.005/0095)
Number of decayed teeth	−0.007 (−0.012/0.003) **	−0.006 (−0.010/−0.001) *	−0.009 (−0.015/−0.004) **	−0.006 (−0.013/0.001)
Number of sextants with bleeding on probing				
0	−0.007 (−0.039/0.025)	0.017 (−0.046/0.081)	−0.002 (−0.047/0.043)	−0.025 (−0.080/0.029)
1	0.008 (−0.054/0.071)	0.001 (−0.029/0.029)	−0.028 (−0.146/0.090)	0.053 (−0.031/0.136)
2	Ref	Ref	Ref	Ref
Number of sextants with dental calculus				
0	0.081 (0.055/0.107) **	0.057 (0.037/0.076) **	0.072 (0.004/0.141) *	0.130 (0.070/0.191) **
1	0.069 (0.028/0.113) **	0.071 (0.054/0.089) **	0.057 (−0.011/0.126)	0.126 (0.006/0.187) **
2	0.069 (0.025/0.113) **	0.032 (−0.033/0.097)	0.050 (−0.028/0.127)	0.152 (0.074/0.230) **
3	0.072 (0.011/0.134) *	0.059 (0.005/0.112) *	0.049 (−0.041/0.139)	0.131 (0.024/0.237) *
4	Ref	Ref	Ref	Ref

* *p* < 0.05, ** *p* < 0.01. The models were adjusted for age, sex and educational level.

## Data Availability

The data that support the finding of this study can be available on request from the corresponding author. The data are not publicly available due to sensitive information that could compromise the privacy of research participants.

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
