# Peer review of "Psychometric Properties of the Brazilian Version of GOHAI among Community-Dwelling Elderly People"

_ijerph, 2022, doi:10.3390/ijerph192214725_

Round 1
Reviewer 1 Report
This study aimed to evaluate the psychometric properties of the translated version of the GOHAI in a representative sample of Brazilian community-dwelling elderly. This article is timely and generally well written and methodologically sound. A number of minor issues arise on reading it and are reported below.
1/ Line 145 page 3: "Participants were interviewed". The GOHAI scale is a self-questionnaire. Can you specify whether the questions were read by the patients or by another person?
2/ Please justify the sample size. Why were 613 elderly people included in the study?
2/ The GOHAI scale is an old scale with some limitations. However, there are new tools such as the IPQ-RDE scale which should be mentioned in the discussion section
Nelson S, Albert JM, Liu Y, Selvaraj D, Curtan S, Ryan K, Pinto A, Ejaz F, Milgrom P, Riedy C. The psychometric properties of a new oral health illness perception measure for adults aged 62 years and older. PLoS One. 2019;10;14(4):e0214082.
Author Response
Reviewer 1
Comments and Suggestions for Authors
This study aimed to evaluate the psychometric properties of the translated version of the GOHAI in a representative sample of Brazilian community-dwelling elderly. This article is timely and generally well written and methodologically sound. A number of minor issues arise on reading it and are reported below.
1/ Line 145 page 3: "Participants were interviewed". The GOHAI scale is a self-questionnaire. Can you specify whether the questions were read by the patients or by another person?
Answer: Thank you for the comment and for asking this clarification. The GOHAI was completed through interviews conducted by a single examiner. The text has been amended to clarify this point.
# Lines 158-159
Added: “The examiner read the items of all questionnaires aloud during the interview, including the GOHAI items.”
2/ Please justify the sample size. Why were 613 elderly people included in the study?
Answer: Thank you for the comment. The justification for the sample size of 613 elderly people has been inserted in the revised version of the manuscript.
# Lines 111-115
Added: “The main study was conducted in 2007 to estimate the prevalence of edentulism in the city of Manaus. The sample size was originally estimated as 807 individuals considering the proportion of 53% of edentulism in the Northern Region of the country [24], with margin of error of 10%, 95% of significance, design effect of 2 and a non-response rate of 20%.”
# Lines 546-547
Added: “24. Brasil, Ministério da Saúde, Coordenação Nacional de Saúde Bucal. Condições de Saúde Bucal da População Brasileira 2002 – 2003 Resultados Principais. Editora MS 2004.”
2/ The GOHAI scale is an old scale with some limitations. However, there are new tools such as the IPQ-RDE scale which should be mentioned in the discussion section
Nelson S, Albert JM, Liu Y, Selvaraj D, Curtan S, Ryan K, Pinto A, Ejaz F, Milgrom P, Riedy C. The psychometric properties of a new oral health illness perception measure for adults aged 62 years and older. PLoS One. 2019;10;14(4):e0214082.
Answer: Thank you for the suggestion. The discussion has been revised and the information about new instruments to assess oral health perceptions of elderly people, such as the IPQ-RDE scale, have been mentioned in the text.
# Lines 460-470
Added: “The importance of measuring OHRQoL is well-established and relies on the shift from traditional clinical measures to assessing the different functional, emotional, and social aspects of oral health to establish realistic treatment objectives and outcomes [5]. Yet, OHRQoL measures for elderly people might be considered narrow since they essentially evaluate the impact of oral health conditions [6]. The GOHAI is particularly limited than other OHRQoL measures due to greater emphasis on functional limitations, pain and discomfort and less discriminant validity when compared with OHIP-14 [37]. Recently, new questionnaires have been developed to assess elderly’s perception of the chronicity of their underlying oral disease and should be considered in future studies, such as the Illness Perception Questionnaire Revised for Dental Use in Older/Elder Adults (IPQ-RDE), [38].”
# Lines 573-578
Added: “37. Osta, N.E.; Haddad, E.; Fakhouri, J.; Saad, R.; Osta, L.E. Comparison of psychometric properties of GOHAI, OHIP-14 and OHIP-EDENT as measures of oral health in complete edentulous patients aged 60 years and more. Qual. Life Res. 2021, 30, 1199–1213. https://doi.org/10.1007/s11136-020-02709-w.
- Nelson, S.; Albert, J.M.; Liu, Y.; Selvaraj, D.; Curtan, S.; Ryan, K.; Pinto, A.; Ejaz, F.; Milgrom, P.; Riedy, C. The psychometric properties of a new oral health illness perception measure for adults aged 62 years and older. PLoS One. 2019, 14, e0214082. https://doi.org/10.1371/journal.pone.0214082.”
Reviewer 2 Report
Dear authors!
The presented article is interesting and valuable, but some questions have arisen so improvements are required.
Comments are attached in the file.

Author Response
Reviewer 2
- Line number 33.
Please select and specify keywords using this resource: https://meshb.nlm.nih.gov/
Answer: Thank you for the suggestion. The keywords have been replaced using MeSH terms.
# Line 33
Removed: “Keywords: Psychometric properties; GOHAI; Oral health-related quality of life; Elderly people”
Added: “Keywords: Psychometrics; Oral health; Quality of life; Aged”
- Dear authors, I could not find the information describing the Cross-Cultural Adaptation Process to the Portuguese language in detail both in the text of the manuscript and in the bibliographic references [17-19, 21]. Perhaps this is due to the fact that the submitted bibliographic references are not in English. The presented study is essential. However, it is an additional stage after the successful completion of the main one (adaptation). Please, expand this issue in the text of the manuscript. In particular, if Cross-Cultural Adaptation was not carried out, can this affect the reliability of the received results? Otherwise, complete the text with a brief description of this process.
Answer: Thank for the thoughtful comment. The Brazilian version of the GOHAI was not cross-culturally adapted and this may explain the findings of our study. This information has been inserted in the revised version of the manuscript.
# Lines 85-86
Removed: “The GOHAI instrument was translated into Portuguese to assess the relationship of ...”
Added: “The GOHAI instrument was not cross-culturally adapted into Portuguese. However, the translated version of GOHAI into Portuguese was used to assess the relationship of …”
# Lines 434-438
Added: “The above-mentioned findings may have occurred because the cross-cultural adaptation of GOHAI for Brazilian elderly people was not conducted yet. Instead, the translated version of GOHAI into Portuguese was used in previous research but the psychometric characteristics of the instrument was not evaluated until now [17-21].”
- Line number 492
The DOI in the bibliographic reference is incorrect, since it belongs to the reference #21.
Answer: Thank you for identifying this problem. The DOI of the reference Cardoso et al 2011 has been replaced for the correct one.
# Line 543
Removed: “doi: 10.1111/j.1741-2358.2010.00417.x.”
Added: “https://doi.org/10.1590/S1415-790X2011000100012.”
- Line number 95–101
The bibliographic source on the basis of which you have chosen the method of forming and determining the sample size describes the parameters, but there is no information about the calculation method (formula, method, computer program?). This information is necessary for the repetition possibility of the presented research. Please specify why the calculation of the sample size is based on data for 2007?
Answer: Thank you for the comment. The sample size calculation has been inserted in the revised version of the manuscript. The data was collected in 2007 and thus the sample size calculation considered this year. This information has been inserted in the methods section.
# Lines 111-115
Added: “The main study was conducted in 2007 to estimate the prevalence of edentulism in the city of Manaus. The sample size was originally estimated as 807 individuals considering the proportion of 53% of edentulism in the Northern Region of the country [24], with margin of error of 10%, 95% of significance, design effect of 2 and a non-response rate of 20%.”
# Lines 546-547
Added: “24. Brasil, Ministério da Saúde, Coordenação Nacional de Saúde Bucal. Condições de Saúde Bucal da População Brasileira 2002 – 2003 Resultados Principais. Editora MS 2004.”
- Line number 110
In a study by Cardoso E.M. et al. (2011) is indicated the same ethics committee protocol number (0234.0.115.000-07). Tell me, please, has the presented study (GOHAI) been declared within the framework of this protocol? Each study should be examined by the ethics committee separately. –
Answer: The data collection procedures reported in the research protocol of the main study approved by the by the Ethics Committee of the Federal University of Amazonas (Protocol No. 0234.0.115.000-07) included the interviews and clinical examinations to collect the sociodemographic characteristics, dental clinical data and subjective measures, including the GOHAI. So, the findings reported in this manuscript were based in the data that was approved by the ethics committee.
- Please add the «Study Limitations» section.
Answer: The limitations of the study were addressed in the last paragraph of the Discussion section (see lines 454-459).